# Effect of Microbial Transglutaminase Treatment on the Techno-Functional Properties of Mung Bean Protein Isolate

**DOI:** 10.3390/foods12101998

**Published:** 2023-05-15

**Authors:** Su-Hyeon Moon, Seong-Jun Cho

**Affiliations:** 1Department of Food Science and Biotechnology, Kangwon National University, Chuncheon 24341, Republic of Korea; anstngus0@kangwon.ac.kr; 2ALT LAB Co., Ltd., Chuncheon 24341, Republic of Korea

**Keywords:** microbial transglutaminase, mung bean protein isolate, electrophoresis, techno-functional properties, texture analysis

## Abstract

The purpose of this study was to investigate the improvement in techno-functional properties of mung bean protein isolate (MBPI) treated with microbial transglutaminase (MTG), including water- and oil-holding capacity, gelling properties, and emulsifying capacity. MBPI dispersions were incubated with MTG (5 U/g of protein substrate) at 45 °C with constant stirring for 4 h (MTM4) or 8 h (MTM8). Sodium dodecyl sulfate-polyacrylamide gel electrophoresis showed that MTG treatment for different durations increased the amount of high-molecular-weight proteins in MBPI, and most of the cross-linking by MTG was terminated at 8 h. Improved water-holding capacity, gelling properties, emulsifying capacity, and stability were observed after MTG treatment, and decreased protein solubility and surface hydrophobicity were observed. Furthermore, the texture of the heat-induced gels made from MTG-treated MBPI was evaluated using a texture analyzer. MTG treatment increased the hardness, gumminess, chewiness, and adhesiveness of the heat-induced gels. Field-emission scanning electron microscopy demonstrated the enhanced hardness of the gels. This research reveals that MTG-catalyzed cross-linking may adjust the techno-functional properties of MBPI, allowing it to be used as a soy protein alternative in food products, such as plant-based and processed meats.

## 1. Introduction

Proteins are essential nutrients for humans. Since ancient times, meat has been the most easily accessible source of protein; however, world population growth, food shortages, livestock disease, ethical consumption, and the increased uptake of veganism means meat intake has gradually reduced, and the consumption of vegetable protein is continuously increasing [1]. Among plant proteins, soybean protein is economical and contains rich, beneficial nutrients, such as essential amino acids and fatty acids [2]. However, trypsin inhibitors, anti-nutrients found in soybean protein, reduce its nutritional value by inhibiting the activity of proteolytic enzyme trypsin [3]. Soybeans are one of eight major allergenic foods, which account for 90% of food allergies [4], and approximately 0.5% of the general population is allergic to soybean products [5]. Mung bean protein has received considerable attention as a suitable replacement for soybean protein.

The mung bean (*Vigna radiata* L.) is a leguminous plant widely cultivated in Asia, including Korea, India, and China because of its nutritional value, such as a high protein content of 20–27%, low fat content, and richness in vitamins B and C, manganese, iron, and calcium. It also contains sufficient amounts of all amino acids, including methionine and cystine, except tryptophan [6]. Mung bean protein has an essential amino acid content comparable to that of soybean protein, and its essential amino acid composition is significantly similar to that of soybean protein and the FAO/WHO reference protein [7]. Additionally, its trypsin inhibitor activity, which can reduce nutritional value, is reportedly lower than that of soybean protein [8]. However, MBPI has inferior techno-functional properties compared to soybean protein, including its water- and oil-holding capacity, emulsifying capacity and stability, and gelling properties [9]. Therefore, various modification methods for solving this problem have been studied [10,11]. However, there has been no extensive exploration of the techno-functional properties of microbial transglutaminase (MTG)-treated MBPI and a texture analysis of heat-induced gels made from MTG-treated MBPI has not yet been reported.

MTG from *Streptovericillium mobaraense* is mainly used to enhance protein techno-functional properties. It is an enzyme that catalyzes the acyl-transfer reaction between the γ-carboxamide group of glutamine residues and diverse primary amines, including the ε-amino group of lysine residues; this acyl-transfer reaction forms an ε-(γ-glutamyl)lysine isopeptide bond, causing changes in the molecular weight, molecular structure, and surface hydrophobicity of the protein. As these changes can enhance the techno-functional properties of proteins, they are widely applied in various food industries such as dairy and bread products and plant-based meat [12]. Additionally, it can be utilized not only in the food industries but also in various applications such as encapsulation and film [13,14]. Therefore, MTG might be used to enhance the inferior techno-functional properties of MBPI.

In this study, sodium dodecyl sulfate-polyacrylamide gel electrophoresis (SDS-PAGE) and surface hydrophobicity analyses were performed to analyze structural changes in MBPI after MTG treatment. The effect of MTG on solubility, which is an important functional property of a protein, and its techno-functional properties, including water- and oil-holding capacity, emulsion capacity and stability, foaming capacity and stability, and the least gelling concentration were measured. To analyze the effect of MTG treatment on the texture profiles of gels from MBPI, heat-induced gels made from MTG-treated MBPI were analyzed using a texture analyzer and field-emission scanning electron microscope (FE-SEM).

## 2. Materials and Methods

### 2.1. Raw Materials and Chemicals

Mung bean (*Vigna radiata* L.) flour was purchased from a local market in South Korea. Commercial MTG (enzyme activity 150 U/g protein) was purchased from DYNE SOZE (Yongin-si, Gyeonggi-do, Republic of Korea) and used in its original form without further purification. Urea, 8-anilino-1-naphthalene sulfonic acid (ANS), and sodium dodecyl sulfate (SDS) were purchased from Sigma-Aldrich Co. (St. Louis, MO, USA). All other reagents and chemicals used in this study were of analytical grade.

### 2.2. Preparation of Mung Bean Protein Isolate (MBPI)

MBPI was prepared according to a previously reported method with minor modifications [15]. Briefly, the mung bean flour was dispersed in double-distilled water (DDW) at a ratio of 1:8 (*w*/*v*). The dispersion was adjusted to pH 7.0 with 1 M NaOH, then stirred for 1 h at room temperature (RT). After extraction, the dispersion was centrifuged at 3200× *g* for 30 min. The supernatant was collected, adjusted to pH 4.5 with 1 M HCl, and centrifuged at 3200× *g* for 30 min. Subsequently, the protein precipitate was neutralized to pH 7.0 with 1 M NaOH. The neutralized mung bean protein was spray-dried and stored at RT. The protein content was 80.7 ± 0.3% (dry basis), as determined using the Kjeldahl method with Kjeltec8200 (Foss, Hillerod, Denmark) using a nitrogen-to-protein conversion factor of 6.25.

### 2.3. Preparation of MTG-Treated MBPI

MTG treatment of MBPI was carried out according to a previously reported method [16] with minor modifications. MBPI (50 g) was dispersed in 1000 mL DDW for 1 h at RT. Next, sodium azide was added at a final concentration of 0.01% (*w*/*v*) to inhibit microbial growth. MBPI dispersions were incubated with MTG (5 U/g of protein substrate) at 45 °C with constant stirring. The MBPI dispersion was sampled at 2 h intervals for SDS-PAGE, and the dispersion was allowed to react for 4 h (MTG-treated MBPI, MTM4) or 8 h (MTM8). Finally, the MTG reaction was terminated by the addition of NH_4_Cl at a final concentration of 10 mM. For the control MBPI (CTL), all details remained identical for 8 h, and no enzyme was added. MTM4, MTM8, and CTL were lyophilized and milled.

### 2.4. Effects of MTG Treatment on MBPI

#### 2.4.1. Electrophoresis

Sodium dodecyl sulfate-polyacrylamide gel electrolysis (SDS-PAGE) was carried out according to a previously reported method with minor modifications [17]. A 10% acrylamide separating gel and 5% acrylamide stacking gel were used in this study. The samples were diluted 10-fold with DDW, and mixed with 5 × SDS-PAGE loading buffer (62.5 mM β-mercaptoethanol, 10% SDS, 50% glycerol, 0.1% bromophenol, 250 mM Tris-HCl, pH 6.8), heated at 90 °C for 10 min and cooled at RT. Finally, the mixture was centrifuged at 12,000× *g* for 5 min. Next, 10 μL loading sample supernatant and 5 μL Dyne unstained protein marker were loaded onto each lane of the gel with 10 × tricine running buffer diluted 10-fold with DDW. Electrophoresis was performed at 200 V for 40 min until the tracking dye reached the bottom of the gel, followed by staining with Coomassie Blue R250 for 1 h. The gel was de-stained using a de-staining solution (8:1:1, DDW: methanol: acetic acid). The percentage of proteins with a molecular weight of 70 kDa or more was confirmed using ImageJ software (Version 1.8.0, NIH, Bethesda, MD, USA).

#### 2.4.2. Surface Hydrophobicity (H_0_)

Surface hydrophobicity was determined according to a previously reported method with minor modifications [18]. MBPI, CTL, MTM4, and MTM8 were dispersed in DDW for 1 h at RT to obtain 30 mg/mL concentration. The protein dispersions were centrifuged at 3200× *g* for 30 min, and samples were prepared by filtering the supernatants through filter paper (F1001, CHMLAB, Terrassa-Barcelona, Spain). The samples were diluted with 10 mM sodium phosphate buffer (pH 7.0) to give five gradient concentrations ranging from 0.15–0.75 mg/mL. Next, 1 mL each sample was added to 5 μL ANS reagent (8.0 mM 1-anilino-8-naphthalene sulfonate in dimethyl sulfoxide) and vortexed for 5 s. The samples were kept in the dark for 10 min before 200 μL each sample was pipetted into a 96-well plate. Fluorescence intensity was measured using a fluorescence spectrophotometer (SpectraMax i3, Molecular Devices, San Jose, CA, USA) at excitation and emission wavelengths of 390 and 470 nm, respectively. Surface hydrophobicity was defined as the initial slope of the linear regression curve between fluorescence intensity and protein concentration.

#### 2.4.3. Protein Solubility

Protein solubility was determined according to a previously reported method with minor modifications [19]. MBPI, CTL, MTM4, and MTM8 were dispersed with DDW and adjusted to pH 2.0–12.0 with 1.0 M HCl or 1.0 M NaOH for 30 min to obtain a 10 mg/mL concentration. The protein dispersions were centrifuged at 3200× *g* for 30 min, and the supernatants were filtered through filter paper (F1001, CHMLAB, Terrassa-Barcelona, Spain). The supernatants and dispersions for the total protein content were diluted 4-fold with a solution containing SDS and urea and allowed to react for 30 min. The final SDS and urea concentrations were 1.5 M and 0.5%, respectively. The protein content of the samples was determined according to the bicinchoninic acid (BCA) assay using a Pierce BCA Protein Assay Kit (Thermo Fisher Scientific, Waltham, MA, USA), and bovine serum albumin was used as a standard. Solubility was presented as a percentage of the protein content in the supernatant relative to the total protein content.

### 2.5. Techno-Functional Properties

#### 2.5.1. Water- and Oil-Holding Capacity (WHC/OHC)

WHC and OHC were determined according to a previously reported method with minor modifications [20]. After the weight of the centrifuge tube was measured, 0.1 g of MBPI, CTL, MTM4, and MTM8 were dispersed in 1 mL of DDW or soybean oil, vortexed for 1 min, and centrifuged at 12,000× *g* for 5 min. The supernatant was removed, and the centrifuge tube containing the precipitate was weighed. WHC and OHC are expressed in grams of water or oil absorbed per gram of protein isolate, respectively. WHC and OHC were calculated using the following equations:(1)$$WHC\;or\;OHC(g/g)=\frac{W_{2}-W_{1}}{W_{0}}$$
where *W*_1_ is the weight of the centrifuge tube containing the dry sample, *W*_2_ is the weight of the centrifuge tube after decantation with oil or water, and *W*_0_ is the weight of the sample.

#### 2.5.2. Emulsifying Capacity and Stability (EC/ES)

EC and ES were measured according to a previously reported method with minor modifications [20]. After dispersing 2 g of MBPI, CTL, MTM4, or MTM8 in 25 mL of DDW, 25 mL of soybean oil was added and homogenized at 20,000 rpm for 2 min using a homogenizer (T25 Digital Ultra-Turrax, IKA, Sutaufen, Germany). A total volume of 40 mL (*V_T_*) emulsion was placed in a 50 mL centrifuge tube and centrifuged at 1500× *g* for 5 min, and the volume (*V_F_*_1_) of the remaining emulsion was measured. After heating the emulsion at 80 °C for 30 min and centrifuging again at 1500× *g* for 5 min, the volume (V_F2_) of the remaining emulsion was measured. ES and EC were calculated using the following equations:(2)$$EC\left(\%,v/v\right)=\frac{V_{F1}}{V_{T}}\times 100$$
(3)$$ES\left(\%,v/v\right)=\frac{V_{F2}}{V_{F1}}\times 100$$
where *V_T_* is the total volume of the emulsion (40 mL), *V_F_*_1_ is the volume of the emulsion remaining after centrifugation, and *V_F_*_2_ is the volume of the emulsion remaining after heating and centrifugation.

#### 2.5.3. Foaming Capacity and Stability (FC/FS)

FC and FS were measured according to a previously reported method with minor modifications [21]. After preparing 1% (*w*/*v*) MBPI, CTL, MTM4, and MTM8 solutions, 100 mL (V_1_) of each solution was prepared and homogenized at 18,000 rpm for 5 min using a homogenizer (T25 Digital Ultra-Turrax, IKA). The volume (*V*_0_) of the produced foam was measured and left for 60 min, after which the volume (*V*_60_) of the foam was measured. The FS and FC were calculated using the following equations:(4)$$FC\left(\%,v/v\right)=\frac{\left(V_{0}-V_{1}\right)}{V_{1}}\times 100$$
(5)$$FS\left(\%,v/v\right)=\frac{\left(V_{60}-V_{1}\right)}{\left(V_{0}-V_{1}\right)}\times 100$$
where *V*_1_ is the initial volume of the solution (100 mL), *V*_0_ is the volume of the foam after homogenization, and *V*_60_ is the volume of the foam remaining after 60 min.

#### 2.5.4. Least Gelling Concentration (LGC)

LGC was measured according to a previously reported method with minor modifications [22]. After preparing 10–30% (*w*/*w*) MBPI, CTL, MTM4, and MTM8 solution with 75 mM potassium phosphate buffer (pH 7.6) in a glass vial, the mixture was heated at 90 °C for 30 min with the glass vial lid closed. After heating, the mixture was cooled at RT for 1 h and stored at 4 °C overnight. The least gelling concentration was defined as the concentration at which the sample did not slip or fall from the inverted test tube.

### 2.6. Characterization of Heat-Induced Protein Gel

#### 2.6.1. Texture Profile Analysis (TPA)

After preparing 23% (*w*/*w*) MBPI, CTL, MTM4, and MTM8 solution with buffer in a 50 mL conical tube, the mixture was heated at 90 °C for 30 min with the conical tube lid closed. After heating, the mixture was cooled at RT for 1 h and stored at 4 °C overnight. The prepared gel was uniformly cut into a cylindrical shape with a height of 20 mm and a diameter of 25 mm then subjected to TPA using a texture analyzer (TA.XTplus100C, Stable Micro System, Godalming, UK). Measurement conditions were determined according to a previously reported method with minor modifications [23]. The parameters were set as follows: a cylinder probe with a diameter of 36 mm was used, with the test mode set to compression; the pre-test speed was 5.0 mm/s, the test speed was 1.0 mm/s, the post-test speed was 5.0 mm/s, the distance was 4.0 mm, and the trigger force was 5 g; the trigger type was set to auto.

#### 2.6.2. Field Emission Scanning Electron Microscopy (FE-SEM)

After preparing 23% (*w*/*w*) MBPI, CTL, MTM4, and MTM8 solution with buffer in a glass vial, the mixture was heated at 90 °C for 30 min with the glass vial lid closed. After heating, the mixture was cooled at RT for 1 h and stored at 4 °C overnight. The gels were then lyophilized and crushed. The dried samples were then sputter-coated with platinum (Cressington sputter coater 208HR, Cressington Scientific Instruments Ltd., Watford, UK) and the microstructure was observed using FE-SEM (JSM-7900F, JEOL Ltd., Akishima, Tokyo, Japan) at ×300 magnification.

### 2.7. Statistical Analysis

All the experiments were performed in triplicate, and results are expressed as mean ± standard deviation. Statistical analyses were conducted using SPSS software (version 26.0, SPSS, Inc., Chicago, IL, USA). A one-way analysis of variance (ANOVA) followed by Duncan’s multiple range test for post hoc comparisons were used to determine significant differences among groups. Statistical significance was set at *p* < 0.05.

## 3. Results and Discussion

### 3.1. Electrophoresis

The SDS-PAGE patterns and the ratios of protein subunits over 70 kDa after different MTG reaction durations are shown in Figure 1A,B, respectively. The MBPI profile showed major bands at 50–70 kDa and ~25 kDa (Figure 1A, lane MBPI). The major bands of 50–60 kDa, approximately 25 kDa, correspond to subunits of vicilin-like 7S (Figure 1A, a), and the 68 kDa band corresponds to legumin-like 11S (Figure 1A, b) [24,25]. The bands in the MBPI lane and the 0 h lane showed similar patterns, indicating that MTG inactivation by NH4Cl was effective and that MTG addition did not affect the SDS-PAGE pattern (Figure 1A, lane 0 h). After more than 2 h, the intensity of the band below 70 kDa, including the major band, gradually decreased with increasing MTG treatment duration, and a new band was formed above 70 kDa (Figure 1A, p). These decreases and the formation of the band indicate that protein subunits, such as vicilin-like 7S and legumin-like 11S, are cross-linked by MTG, increasing their molecular weight. In addition, a new band was formed at the top of the stacking gel, which indicates that polymerized proteins treated with MTG did not pass through the stacking gel due to increased molecular weight [26]. There was a significant difference in the ratio of protein subunits over 70 kDa between MBPI treated with MTG for 0, 2, 4, and 6 h; however, no significant difference was observed between MBPI treated for 8 h or more (Figure 1B). MTG treatment increased the molecular weight of MBPI through protein subunit crosslinking, and most of the reactions induced by MTG were terminated at 8 h.

### 3.2. Surface Hydrophobicity (H_0_)

The surface hydrophobicity (H_0_) of a protein indicates the relative content of hydrophobic residues on its surface, and changing the protein tertiary structure by MTG affects the H_0_ value [27]. The surface hydrophobicity of CTL (2.25 ± 0.11 × 10^5^) was significantly higher than that of MBPI (1.48 ± 0.10 × 10^5^) (Figure 2). This increase in the surface hydrophobicity of CTL was due to its denaturation via lyophilization after spray-drying; lyophilization increases the surface hydrophobicity of MBPI [15]. The surface hydrophobicity of MTM4 (1.80 ± 0.07 × 10^5^) and MTM8 (1.40 ± 0.07 × 10^5^) was significantly lower than that of CTL, and as the MTG treatment duration increased, the surface hydrophobicity significantly decreased. MTG treatment changed the tertiary structure of the protein via crosslinking and reduced the exposure of hydrophobic groups on the protein surface [28]. MTG treatment may reduce the surface hydrophobicity of proteins by partial deamination of glutamine and ε-amino groups [29]. The change in tertiary structure and the balance of hydrophilic and hydrophobic groups by MTG may affect techno-functional properties, such as emulsifying properties and water- and oil-holding capacity [16].

### 3.3. Protein Solubility

Water solubility is one of the most important functional properties of proteins used in the food industry. The formation of high-molecular-weight proteins by MTG treatment changes protein solubility, and these changes may affect functional properties, such as gelling, emulsifying, and foaming properties [30]. MBPI, CTL, MTM4, and MTM8 exhibited similar protein solubility patterns, with minimum solubility at pH 4.0–5.0. The protein solubility progressively increased below and above pH 4.0–5.0 except for a slight decrease at pH 9.0–10.0 (Figure 3). Except for pH 4.0–5.0, the protein solubility of CTL was slightly decreased compared to MBPI, and that of MTM4 and MTM8 was significantly decreased. Protein solubility was affected by surface hydrophobic groups, and the surface hydrophobicity of CTL was increased compared to that of MBPI, resulting in decreased CTL protein solubility. Although the surface hydrophobicity of MTM4 and MTM8 was lower than that of CTL, protein solubility was not improved. This suggests that the effect of cross-linking, changing the protein tertiary structure, and forming high-molecular-weight insoluble proteins is a more critical parameter than that of decreasing surface hydrophobicity. These decreases in solubility are in agreement with previous results observed for MTG-treated walnut protein isolates and vicilin-rich kidney protein isolates [31,32].

### 3.4. Techno-Functional Properties

#### 3.4.1. Water- and Oil- Holding Capacity (WHC/OHC)

WHC is the ability to absorb and retain water molecules in a protein matrix, which is determined by various physicochemical factors, such as the molecular structure and surface charge of amino acids. The WHC of MTM4 (3.74 ± 0.12 *g*/*g*) and MTM8 (4.08 ± 0.08 *g*/*g*) significantly increased with the highest value being that of MTM8, but there was no significant difference between CTL (2.00 ± 0.05 *g*/*g*) and MBPI (1.90 ± 0.03 *g*/*g*) (Table 1). Although there was no statistically significant difference in WHC between MTM8 and MTM4, that of MTM8 (4.08 ± 0.08 *g*/*g*) was slightly higher than MTM4 (3.74 ± 0.12 *g*/*g*). The cross-linking formed by MTG treatment changed the tertiary structure of the protein, which can potentially physically retain water in the protein matrix. These increases in WHC agree with previous results for MTG-treated wheat, barley, soybean, and whey protein [33,34].

OHC is the ability to absorb and retain oil molecules in a protein matrix, and it is affected by various physicochemical factors, mainly surface hydrophobicity [35]. The OHC of CTL (2.28 ± 0.14 *g*/*g*) increased to the highest value, but there was no significant difference between that of MTM4 (2.03 ± 0.12 *g*/*g*), MTM8 (2.10 ± 0.04 *g*/*g*), and MBPI (2.16 ± 0.07 *g*/*g*) (Table 1). The OHC of CTL slightly increased compared to MBPI due to the increase in surface hydrophobicity (Figure 2); this may not be because of MTG treatment but rather the difference in drying method. MTG treatment did not significantly improve OHC, perhaps because it reduces the surface hydrophobicity of MBPI; as a result, MTG treatment did not affect OHC in this study.

#### 3.4.2. Emulsifying Capacity and Stability (EC/ES)

Proteins with hydrophobic and hydrophilic regions act as emulsifiers. The emulsifying ability of a protein is mainly affected by its molecular structure, the ratio of hydrophilicity to hydrophobicity of its amino acids, and its molecular weight [36,37]. The EC of MTM4 (66.7 ± 3.6%, *v*/*v*) significantly increased, with MTM8 (68.8 ± 0.0%, *v*/*v*) reaching the highest value, but there was no significant difference between that of CTL (56.3 ± 0.0%, *v*/*v*) and MBPI (54.2 ± 3.6%, *v*/*v*) (Table 1). Generally, an increase in the surface hydrophobicity of a protein enhances its emulsifying properties because it is a major factor in adsorbing proteins at oil/water interfaces [38]. However, in the case of MTG treatment, although the surface hydrophobicity of MBPI decreased (Figure 2), its emulsifying ability improved. This was attributed to the fact that for proteins with high solubility, the proportion of surface hydrophobicity is a significant factor in their emulsification parameters, whereas for proteins with low solubility, solubility is a more critical parameter. A decrease in protein solubility can also result in the formation of larger protein aggregates or complexes, which can provide a thicker and more stable oil/water interface, further enhancing their emulsifying properties [28]. Additionally, MTG treatment formed cross-linking, increasing the molecular weight of MBPI (Figure 1). This network traps and stabilizes oil droplets within the emulsion, preventing them from coalescing or separating from the aqueous phase. They can also increase the viscosity and elasticity of the emulsion, resulting in improved stability and reduced coalescence of oil droplets [39]. These increases in EC agree with previous results observed for MTG-treated fava bean protein isolates and soy protein [16,40]. There were no significant differences between the ES of any groups; this shows that MTG treatment did not affect ES in this study.

#### 3.4.3. Foaming Capacity and Stability (FC/FS)

Protein foam is an important component in the food industry because it is key for the production of many food products with unique textures and sensory attributes. A protein foam is typically formed when a protein solution is stirred or whipped in the presence of water and air. This foam can be used as a base for many types of food, including meringues, cakes, mousses, and whipped creams. Although there was no statistically significant difference between the FC of any of the groups, those of CTL (36.7 ± 2.9%, *v*/*v*), MTM4 (36.7 ± 2.9%, *v*/*v*), and MTM8 (38.3 ± 2.9%, *v*/*v*) were slightly lower than that of MBPI (40.0 ± 0.0%, *v*/*v*) (Table 1). In this study, this decrease was not affected by MTG treatment, but by differences in the drying method. These non-significant differences in the FC of MTG treatment groups agree with previous results observed for MTG-treated wheat gluten [29]. The FS of CTL (27.4 ± 2.1%, *v*/*v*) significantly decreased to the lowest value compared to that of MBPI (54.2 ± 7.2%, *v*/*v*); the FS of MTM4 (45.2 ± 4.1%, *v*/*v*) and MTM8 (39.3 ± 3.1%, *v*/*v*) significantly increased compared to CTL (Table 1). The decrease in FS of CTL was likely due to the difference in drying method, while MTG treatment potentially increased the FS of MTM4 and MTM8 compared to CTL. MTG treatment significantly increased the molecular weight of MBPI (Figure 1), which may improve FS. The molecular weight of proteins can affect their foaming properties because larger proteins have a greater number of surface-active amino acid residues, such as lysine, arginine, and histidine, which can reduce the surface tension of the gas-liquid interface and stabilize foam bubbles. However, the relationship between the protein molecular weight and foaming ability is not always straightforward because protein foaming properties are determined by various factors, such as surface hydrophobicity, molecular flexibility, and solution viscosity; the effect of MTG treatment on foaming properties needs to be further investigated [41].

#### 3.4.4. Least Gelling Concentration (LGC)

Protein gels are formed when proteins denature and aggregate, forming a three-dimensional network that traps water and other components. Protein gels are important in the food industry because they are versatile ingredients that provide a range of functional benefits and are used in the production of many different food products. The LGC of MTM4 (12.7 ± 0.6%, *w*/*w*) and MTM8 (12.3 ± 0.6%, *w*/*w*) significantly decreased compared to MBPI (22.3 ± 0.6%, *w*/*w*) and CTL (22.3 ± 0.6%, *w*/*w*), with the LGC of MTM8 being the lowest (Table 1). LGC is a key indicator of protein gel-forming abilities; a lower LGC indicates better gelling properties. The main mechanisms of protein gelation are network formation, including non-covalent interactions, and cross-linking, including covalent bond formation between protein molecules. MTG catalyzes the formation of a covalent bond between the γ-carboxyalkyl group of the deamidated glutamic acid residue and a primary amine group of a nearby lysine residue or a polyamine molecule, such as spermidine. In this study, the ε-(γ-Glu)Lys cross-linked by MTG formed a network capable of holding water molecules, increasing the water holding capacity, and improving the gelling properties of MBPI. These increases in gelling properties agree with previous results observed in MTG-treated canola and pea protein isolates [42,43].

### 3.5. Characterization of Heat-Induced Protein Gels

Texture is determined by the arrangement and interactions of protein molecules within a gel network, as well as by the presence of water and other components. The hardness, gumminess, chewiness, and adhesiveness of the MTM4 and MTM8 gels were significantly increased compared to those of the MBPI and CTL gels. Furthermore, there was no significant difference in springiness, cohesiveness, or resilience among any groups (Table 2). The hardness of the MTM8 gel (1907.5 ± 20.2 g) increased over that of the MTM4 gel (1754.6 ± 71.8 g); the increase in MTG treatment duration had a positive effect on the hardness of the heat-induced gel. These increases can be attributed to the formation of a denser, more uniform gel network structure through protein crosslinking, which results in a harder gel [44]. The FE-SEM images of the heat-induced gel show that the MTG-treated gel formed a denser, more uniform gel network structure than untreated gel (Figure 4).

Gumminess is calculated as the product of hardness and cohesiveness, whereas chewiness is calculated as the product of gumminess and springiness. Since there were no statistically significant differences in cohesiveness or springiness between the groups, gumminess and chewiness showed the same tendency as hardness. As the adhesiveness value increased, the force required to overcome the intermolecular adhesive forces decreased. MTG treatment increased adhesiveness, and gels with high hardness tended to have higher adhesiveness values. Therefore, the addition of MTG positively influenced the textural properties of the heat-induced gel, suggesting its potential application for enhancing the textural quality of food products, such as plant-based fish cakes and processed meats.

## 4. Conclusions

This study revealed the structural changes and techno-functional properties of MTG-treated MBPI. MTG treatment increased MBPI molecular weight through cross-linking and decreased its surface hydrophobicity. The ε-(γ-Glu)Lys cross-linked by MTG formed a network capable of holding water molecules, thereby improving the water holding capacity and gelling properties of MBPI. Cross-linking provides a more complex structure with more amino acids and functional groups available for binding to oil droplets. Furthermore, the decrease in solubility provides a thicker and more stable oil/water interface, thus improving the emulsifying properties of MBPI. The hardness of the heat-induced gel prepared from MBPI significantly increased; FE-SEM images of the heat-induced gel showed that MTG-treated gel formed a denser, more uniform gel network structure than the gel without MTG treatment. This research reveals that MTG-catalyzed cross-linking can enhance the water-holding capacity, gelling properties, and emulsifying properties of MBPI, allowing it to be used as a soy protein alternative in food products, such as plant-based and processed meats.

## Figures and Tables

**Figure 1 foods-12-01998-f001:**
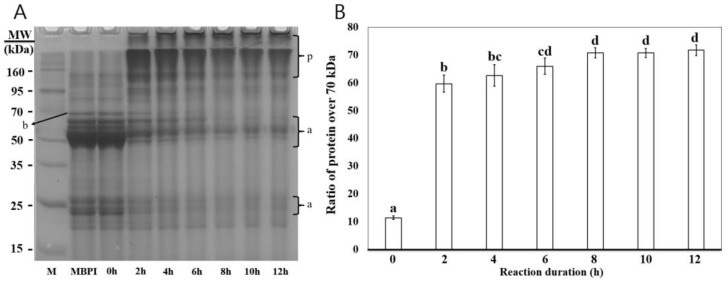
SDS-PAGE patterns (**A**) and the ratio of high molecular weight protein subunits (**B**) of MBPI after different MTG treatment durations. (**A**) SDS-PAGE: lane “M”, molecular weight markers (15–160 kDa); “0–12 h” lanes, MBPI was incubated with MTG for up to 12 h, “p” is new polymeric species formed under MTG treatment, “a” and “b” are vicilin-like 7S and legumin-like 11S, respectively. (**B**) Ratio of proteins > 70 kDa. Different letters on bars indicate a significant difference between groups (*p* < 0.05).

**Figure 2 foods-12-01998-f002:**
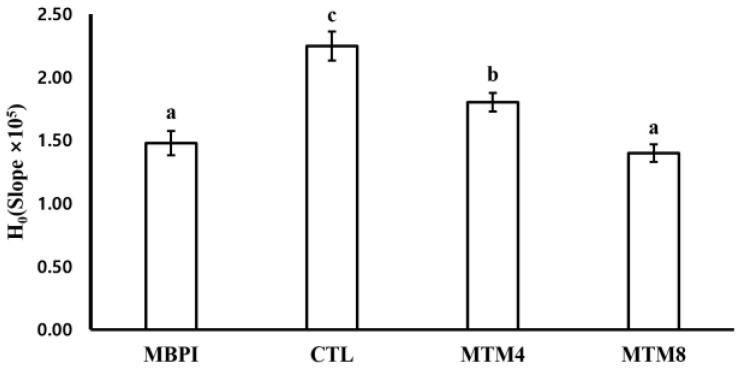
Surface hydrophobicity (H0) of MBPI, CTL, MTM4, and MTM8. The CTL was prepared by adding NH4Cl without MTG. MTM4 and MTM8 were incubated with MTG for 4 and 8 h, respectively. Different letters on bars indicate a significant difference between groups (*p* < 0.05).

**Figure 3 foods-12-01998-f003:**
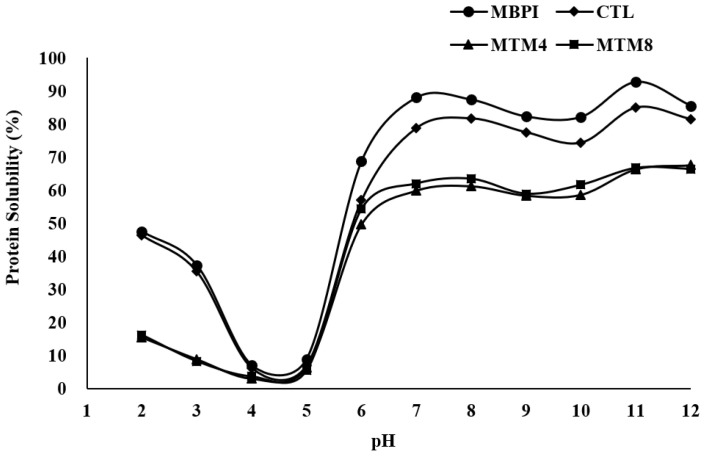
Protein solubility of MBPI, CTL, MTM4, and MTM8. The CTL was prepared by adding NH_4_Cl without MTG. MTM4 and MTM8 were incubated with MTG for 4 and 8 h, respectively.

**Figure 4 foods-12-01998-f004:**
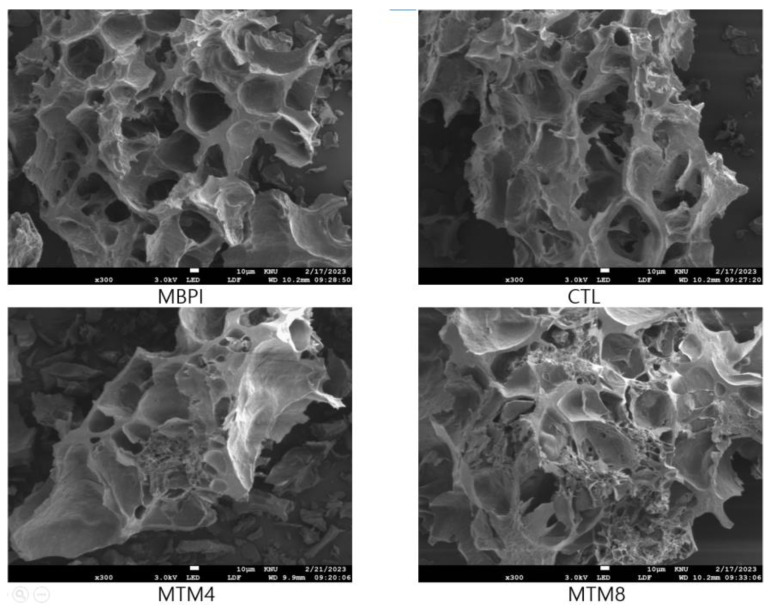
Field emission scanning electron microscopy images of heat-induced gels made from MBPI, CTL, MTM4, and MTM8. The CTL was prepared by adding NH_4_Cl without MTG. MTM4 and MTM8 were incubated with MTG for 4 and 8 h, respectively.

**Table 1 foods-12-01998-t001:** Techno-functional properties of MBPI, CTL, MTM4, and MTM8.

Fraction	Sample			
	MBPI	CTL	MTM4	MTM8
WHC (*g*/*g*)	1.90 ± 0.03 ^a^	2.00 ± 0.05 ^a^	3.74 ± 0.12 ^b^	4.08 ± 0.08 ^b^
OHC (*g*/*g*)	2.16 ± 0.07 ^ab^	2.28 ± 0.14 ^b^	2.03 ± 0.12 ^a^	2.10 ± 0.04 ^ab^
EC (%, *v*/*v*)	54.2 ± 3.6 ^a^	56.3 ± 0.0 ^a^	66.7 ± 3.6 ^b^	68.8 ± 0.0 ^b^
ES (%, *v*/*v*)	94.6 ± 1.1 ^a^	94.5 ± 5.1 ^a^	97.0 ± 5.3 ^a^	97.3 ± 2.4 ^a^
FC (%, *v*/*v*)	40.0 ± 0.0 ^a^	36.7 ± 2.9 ^a^	36.7 ± 2.9 ^a^	38.3 ± 2.9 ^a^
FS (%, *v*/*v*)	54.2 ± 7.2 ^c^	27.4 ± 2.1 ^a^	45.2 ± 4.1 ^b^	39.3 ± 3.1 ^b^
LGC (%, *w*/*w*)	22.3 ± 0.6 ^b^	22.3 ± 0.6 ^b^	12.7 ± 0.6 ^a^	12.3 ± 0.6 ^a^

The CTL was prepared by adding NH_4_Cl without MTG. MTM4 and MTM8 were incubated with MTG for 4 and 8 h, respectively. WHC, water-holding capacity; OHC, oil-holding capacity; EC, emulsifying capacity; ES, emulsifying stability; FC, foaming capacity; FS, foaming stability; LGC, least gelling concentration. Data are presented as mean ± standard deviation. Different superscript letters in the same row indicate a significant difference (*p* < 0.05).

**Table 2 foods-12-01998-t002:** Textural properties of heat-induced gels made from MBPI, CTL, MTM4, and MTM8.

Parameter	Sample Gel			
	MBPI	CTL	MTM4	MTM8
Hardness (g)	1310.9 ± 50.3 ^a^	1339.9 ± 68.5 ^a^	1754.6 ± 71.8 ^b^	1907.5 ± 20.2 ^c^
Adhesiveness (g·s)	−32.9 ± 3.1 ^a^	−29.8 ± 1.9 ^a^	−16.4 ± 1.4 ^b^	−11.2 ± 1.6 ^c^
Springiness	1.0 ± 0.0 ^a^	1.0 ± 0.0 ^a^	1.0 ± 0.0 ^a^	1.0 ± 0.0 ^a^
Cohesiveness	1.16 ± 0.07 ^a^	1.31 ± 0.06 ^a^	1.08 ± 0.03 ^a^	1.08 ± 0.03 ^a^
Gumminess (g)	1524.54 ± 133.0 ^a^	1513.6 ± 40.1 ^a^	1897.4 ± 85.6 ^b^	2066.1 ± 41.8 ^c^
Chewiness (g)	1518.44 ± 132.5 ^a^	1507.5 ± 40.0 ^a^	1897.1 ± 84.0 ^b^	2059.2 ± 42.6 ^c^
Resilience	0.14 ± 0.00 ^ab^	0.15 ± 0.01 ^bc^	0.13 ± 0.01 ^a^	0.15 ± 0.01 ^c^

The CTL was prepared by adding NH_4_Cl without MTG. MTM4 and MTM8 were incubated with MTG for 4 and 8 h, respectively. Data are presented as mean ± standard deviation. Different superscript letters in the same row indicate a significant difference (*p* < 0.05).

## Data Availability

The data presented in this study are available on request from the corresponding author.

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
