# Peer review of "Effect of Microbial Transglutaminase Treatment on the Techno-Functional Properties of Mung Bean Protein Isolate"

_foods, 2023, doi:10.3390/foods12101998_

Round 1
Reviewer 1 Report
The manuscript titled ‘Effect of Microbial Transglutaminase Treatment on the 2 Techno-Functional Properties of Mung Bean Protein Isolate' presents a very Interesting study on the modification of techno-functional properties of mung bean protein using MTGase enzyme. The manuscript is well-written and the results are appropriately presented. However, below are some suggestions to make the manuscript better.
1. The English language must be thoroughly revised.
2. Abstract, line 13, Please revise the sentence for better clarity. MBPI4 makes little sense here.
3. Introduction must be Improved by adding some relevant references. Below are some recent and relevant references about the utilization of mung bean protein Isolate
Gulzar, S., Tagrida, M., Patil, U., Ma, L., Zhang, B. & Benjakul, S. (2023). Mung bean protein isolate treated with high-intensity pulsed electric field: Characteristics and its use for encapsulation of Asian seabass oil. Journal of Microencapsulation, 40, 1-21.
Gulzar, S., Nilsuwan, K., Raju, N., Benjakul, S. (2022). Whole wheat crackers fortified with mixed shrimp oil and tea seed oil microcapsules prepared from mung bean protein isolate and sodium alginate. Foods, 11, 202.
4. Please explain how the H0 of MBPI was lower than MTM4, since the crosslinking will further decrease the hydrophobic areas. Moreover, the SDS PAGE results suggest the formation of high molecular weight Insoluble proteins, the opening of protein structure cannot be the reason for the change in H0 which has been mentioned in the protein solubility section. Please revise the section
5. Line 328 ´MTG treatment decreased the surface hydrophobicity of the protein, which has a repulsive force against water (Figure 2), and can increase the surface chemical binding of proteins with water molecules´. the H0 results suggest otherwise. This is somewhat contradictory to the solubility results in section 3.3. The sentence needs better clarity
6. The WHC of the MTM4 and MTM8 was higher than MBPI, this is also contradictory to the solubility results.
7. How about the reconstitution properties of the modified protein? It is necessary that the protein must have high reconstitution properties since it is necessary to dry the protein to powder form. Does it make a better gel after reconstitution? Considering the solubility of the protein Is reduced.

Author Response
We'd like to thank the editor for managing the review of our work, and we'd also like to thank the reviewers for their insightful and constructive remarks.
Please, check the attated file.

Reviewer 2 Report
This manuscript investigated the effect of microbial transglutaminase (MTG) treatment on the structural and functional properties of mung bean protein isolate, with the aim of enhancing the techno-functional properties of bung bean protein including solubility, WHC, OHC, and gel-forming properties, etc. The abstract and conclusion are clear and concise, and the methods are easy to follow. However, the English writing is poor, and this manuscript needs to be edited by someone with expertise in technical writing. In addition, the authors did not explain the results clearly and reasonably. The detailed comments are as follows:
Line 47-49: Could you please find some references to support your opinion here (mung bean protein isolate (MBPI) has inferior techno-functional properties compared to soybean protein)?
Line 50: I would suggest replacing “improving” by “solving”.
Line 86: Please double check the pH condition here. Is it pH 7.0?
Line 104: For the control group (CTL), did you treat it with incubation under 45 ℃? If yes, could you please specify the duration of the incubation period under 45 ℃ for the control group (CTL)?
Line 284-285: Please find reference to support your statement (“MTG treatment changed the tertiary structure of the protein via cross-linking and reduced the exposure of hydrophobic groups on the protein surface”).
10. Line 338-340: The authors stated that the OHC of CTL was slightly increased. Compared to which treatment group was this increase observed? Also, the authors explained that the increase was resulted from the differences in drying methods. What are the drying methods the authors used in this study? Why could the drying methods lead to the difference? These are the same questions for Line 280-281.
Line 358-360: Please find references to support your statement (“Generally, an increase in the surface hydrophobicity of a protein enhances its emulsifying properties because it is a major factor in adsorbing proteins at oil/water interfaces”).
needs to be improved
Author Response
We'd like to thank the editor for managing the review of our work, and we'd also like to thank the reviewers for their insightful and constructive remarks.
Our responses to the reviewers' comments are presented point by point below.
Line 47-49: Could you please find some references to support your opinion here (mung bean protein isolate (MBPI) has inferior techno-functional properties compared to soybean protein)?
Ans) We are thankful for Reviewer's careful reading and agree with this comment, we added the appropriate references on lines 47-49.
Line 50: I would suggest replacing “improving” by “solving”.
Ans) By agreeing with this comment, we revised "improving" in line 50 of the manuscript to "solving".
Line 86: Please double check the pH condition here. Is it pH 7.0?
Ans) We have double-checked based on the feedback from Reviewer, and it is confirmed that mung bean protein has a relatively high solubility even in neutral pH 7.0, making them suitable for extraction.
Line 104: For the control group (CTL), did you treat it with incubation under 45 ℃? If yes, could you please specify the duration of the incubation period under 45 ℃ for the control group (CTL)?
Ans) CTL was indeed incubated at 45°C, and based on the feedback from Reviewer, we have added information on 8 h incubation at line 105.
Line 284-285: Please find reference to support your statement (“MTG treatment changed the tertiary structure of the protein via cross-linking and reduced the exposure of hydrophobic groups on the protein surface”).
Ans) By agreeing with this comment, we added the appropriate references on lines 284-285.
- Line 338-340: The authors stated that the OHC of CTL was slightly increased. Compared to which treatment group was this increase observed? Also, the authors explained that the increase was resulted from the differences in drying methods. What are the drying methods the authors used in this study? Why could the drying methods lead to the difference? These are the same questions for Line 280-281.
Ans) By agreeing with this comment, we added the information that CTL had an increased OHC compared to MBPI in lines 339-340. And The difference according to the drying method is mentioned as "This increase in the surface hydrophobicity of CTL was due to its denaturation via lyophilization after spray-drying; lyophilization increases the surface hydrophobicity of MBPI." in the text. Additionally, section 2.2 states that MBPI was spray-dried. The reason for the difference in surface hydrophobicity depending on the drying method is explained in the referenced paper below and is also cited in the text.
Brishti, F.H.; Chay, S.Y.; Muhammad, K.; Ismail-Fitry, M.R.; Zarei, M.; Karthikeyan, S.; Saari, N. Effects of Drying Techniques on the Physicochemical, Functional, Thermal, Structural and Rheological Properties of Mung Bean (Vigna Radiata) Protein Isolate Powder. Food Res. Int. 2020, 138, 109783, doi:10.1016/j.foodres.2020.109783.
Line 358-360: Please find references to support your statement (“Generally, an increase in the surface hydrophobicity of a protein enhances its emulsifying properties because it is a major factor in adsorbing proteins at oil/water interfaces”).
Ans) By agreeing with this comment, we added the appropriate references on lines 359-360.
Reviewer 3 Report
The effect of MTG treatment on protein isolate is important in the study of protein-based pseudo-meat. This study systematically examines the effects of MTG and the results are well organized. However, a few minor comments should be noted.
1) Line 21-22
Seems like a repeat of lines 15-17.
2) Line 264
The presence of bands on the stacking gel only indicates the presence of very large molecules and does not indicate whether the reaction is over.
3)Line 308
“protein solubility did not improve” should be “protein solubility was not improved”.
Author Response
We'd like to thank the editor for managing the review of our work, and we'd also like to thank the reviewers for their insightful and constructive remarks.
Our responses to the reviewers' comments are presented point by point below.
1) Line 21-22
Seems like a repeat of lines 15-17.
Ans) We are thankful for Reviewer 3's careful reading and agree with this comment, we revised repeated words in line 50 of the manuscript.
2) Line 264
The presence of bands on the stacking gel only indicates the presence of very large molecules and does not indicate whether the reaction is over.
Ans) While we agree with the opinion of Reviewer 3, based on the information provided in lines 263-264 of the manuscript and Figure 1B, it appears that there is no significant change in the proportion of high molecular weight proteins (over 70 kDa) after 8 hours of treatment with MTG. This may suggest that the reaction has reached completion.
3)Line 308
“protein solubility did not improve” should be “protein solubility was not improved”.
Ans) By agreeing with this comment, we revised “protein solubility did not improve” in line 309 of the manuscript to “protein solubility was not improved”.